# Hess Opinions: Socio-economic and ecological trade-offs of flood management – benefits of a transdisciplinary approach

Karl Auerswald[1], Peter Moyle[2], Simon Paul Seibert[1], Juergen Geist[3]

[1]Grassland Science Unit, Technical University of Munich, 85354 Freising, Germany
[2]Department of Wildlife, Fish, and Conservation Biology, Center for Watershed Sciences, University of California, Davis CA 95616, USA
[3]Aquatic Systems Biology Unit, Technical University of Munich, 85354 Freising, Germany

*Correspondence to*: Juergen Geist (geist@wzw.tum.de)

**Abstract.** In light of climate change and growing numbers of people inhabiting riverine floodplains, worldwide demand for flood protection is increasing, typically through engineering approaches such as more and bigger levees. However, the well-documented "levee effect" of increased floodplain use following levee construction or enhancement often results in increased problems, especially when levees fail or are compromised by big flood events. Herein, we argue that there are also unintended socio-economic and ecological consequences of traditional engineering solutions that need to be better considered, communicated and weighed against alternative solutions. Socio-economic consequences include reduced aesthetic and recreational values as well as increased downstream flooding risk and reduced ecosystem services. Ecological consequences include hydraulic decoupling, loss of biodiversity and increased risk of contamination during flooding. In addition, beyond river losses of connectivity and natural riparian vegetation created by levees, changes in groundwater levels and increased greenhouse gas emissions are likely. Because flood protection requires huge financial investments and results in major and persistent changes to the landscape, more balanced decisions that involve all stakeholders and policy makers should be made in the future. This requires a transdisciplinary approach that considers alternative solutions such as green infrastructure and places emphasis on integrated flood management rather than on reliance on technical protection measures.

## 1. Introduction

Flood protection is high on political agendas worldwide, especially given that climate change is projected to increase the frequency, severity, and extent of floods (Milly et al., 2002; Huntington, 2006). In parallel, the size and wealth of human populations have increased and are likely to increase further. Given that most people in temperate and tropical areas live on or are dependent upon floodplains, there are increasing calls for better flood protection, which is often addressed by building more and bigger levees (Opperman et al., 2017).

Recently, Di Baldassarre et al. (2013, 2018) pointed out that construction of flood-control levees may have unintended and undesired socio-economic consequences. They attribute this to the „levee effect" White (1945) whereby, paradoxically, flood control structures might even increase flood risk: Once levees are built to protect assets such as homes, farms, and commercial buildings from flooding, the sense of security they provide results in more assets being located behind the levees. As asset values increase, the perceived need to further improve levees increases as well, particularly when it is realized that levees do not completely prevent flood events but mainly increase the return interval of large floods. This implies that absolute safety cannot be guaranteed and that dramatic failures may occur even if the return period is predicted to be as much as 1000 years. As climate changes and upstream areas and floodplains are used more intensely, return intervals become shorter. Also, levees can and do fail for many reasons (e.g., earthquakes, aging infrastructure, lack of maintenance; Burton and Cutter, 2008). These realities are often not conveyed well to stakeholders who live behind the levees (e.g., Ludy and Kondolf, 2012). Therefore, Di Baldassarre et al. (2018) propose a research agenda to "significantly improve our understanding of the unintended effects of flood protection" with a special emphasis on human behaviour. While this research agenda is important, our experiences show that the levee effect is not the only negative impact of levee construction, but that a whole suite of unintended socioeconomic and ecological consequences within and beyond the river systems are the unavoidable result of levee construction and maintenance (Fig. 1). Dependence on levees and river engineering (e,g,. channelization) is already widespread, especially in Central Europe and the USA (Dynesius and Nilsson, 1994; Nilsson et al., 2005), where levees have been built almost along all larger rivers and where even many of the smallest streams have been subjected to engineering „fixes". We therefore argue that levee construction and river engineering have reached or even exceeded bearable levels and that alternative management approaches are needed. Fortunately, other options are available to meet the multitude of society's demands for river services (Opperman et al., 2017).

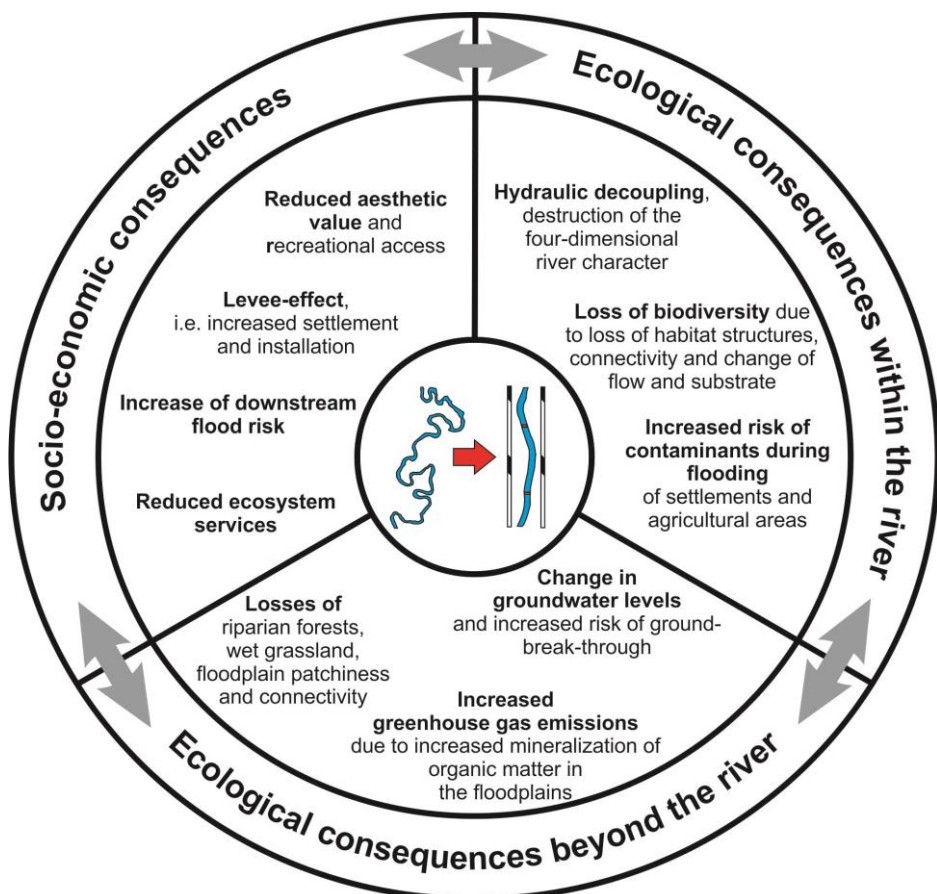

**Figure 1: Unintended consequences of structural flood protection include socio-economic as well as ecological consequences within and beyond the river system.**

## 2. Socio-economic consequences

The levee effect emphasized by Di Baldassarre et al. (2018) is not the only unintended and undesirable socio-economic effect of levees. Levees are usually built in tandem with dams and, in certain cases, retention structures that buffer peak flows (e.g., artificial floodplains or water holding structures) and modified channels. The construction of levees is typically associated with straightening the water course (Fig. 2 a, b, c) and squeezing bank-full river flow into a corset of a larger hydraulic gradient, higher effective flow radius and lower roughness. This inevitably increases flow velocity, as described by the well-established Manning-Gauckler-Strickler relationship (Strickler, 1924). This increase was initially thought to be desirable as it increased riverbed erosion and incision of the river, which reduced required dam heights (Fig. 2 d, detail view) and associated costs. However, the increase in flow velocity is sustained downstream and, according to the fundamentals of fluid mechanics, every increase in flow velocity increases peak flow rate (Sherman, 1932; Bormann et al., 1999). As a consequence, any levee construction that aims at fast and safe drainage increases flood risk downstream, including erosion

and destabilization of river beds and levees. The increase in flood risk downstream by single flood control measures may be small but impacts of catchment-wide river channel changes (e.g., by installing levees, lowering flow bases, and the straightening of river courses) accumulate. Concurrently this can have very large and complex effects on the water balance, particularly on hydrologic extremes (Pattison and Lane, 2012). This may explain why return intervals of floods dramatically

5   decrease over time (Vogel et al., 2011) and can start a new, expensive cycle of levee construction.

While such artificial modifications have generally benefited urban centres and industrial-scale agricultural developments, river-dependent communities and individuals who live downstream of dams and levees have commonly experienced loss of livelihoods, food security, and other factors contributing to their physical, cultural and spiritual well-being (Richter et al., 2010). Unwanted consequences of levee construction include direct impacts on provisioning services (e.g., reduced fish

10   productivity), regulating services (e.g. reduced buffering function of intact floodplains), habitat or supporting services (e.g., decline of connectivity-dependent migratory fishes), as well as reduced cultural services (e.g., reduced aesthetic appeal of engineered versus natural river courses). These services are difficult to express in monetary terms but their actual economic values are likely to be high (Opperman et al. 2017).

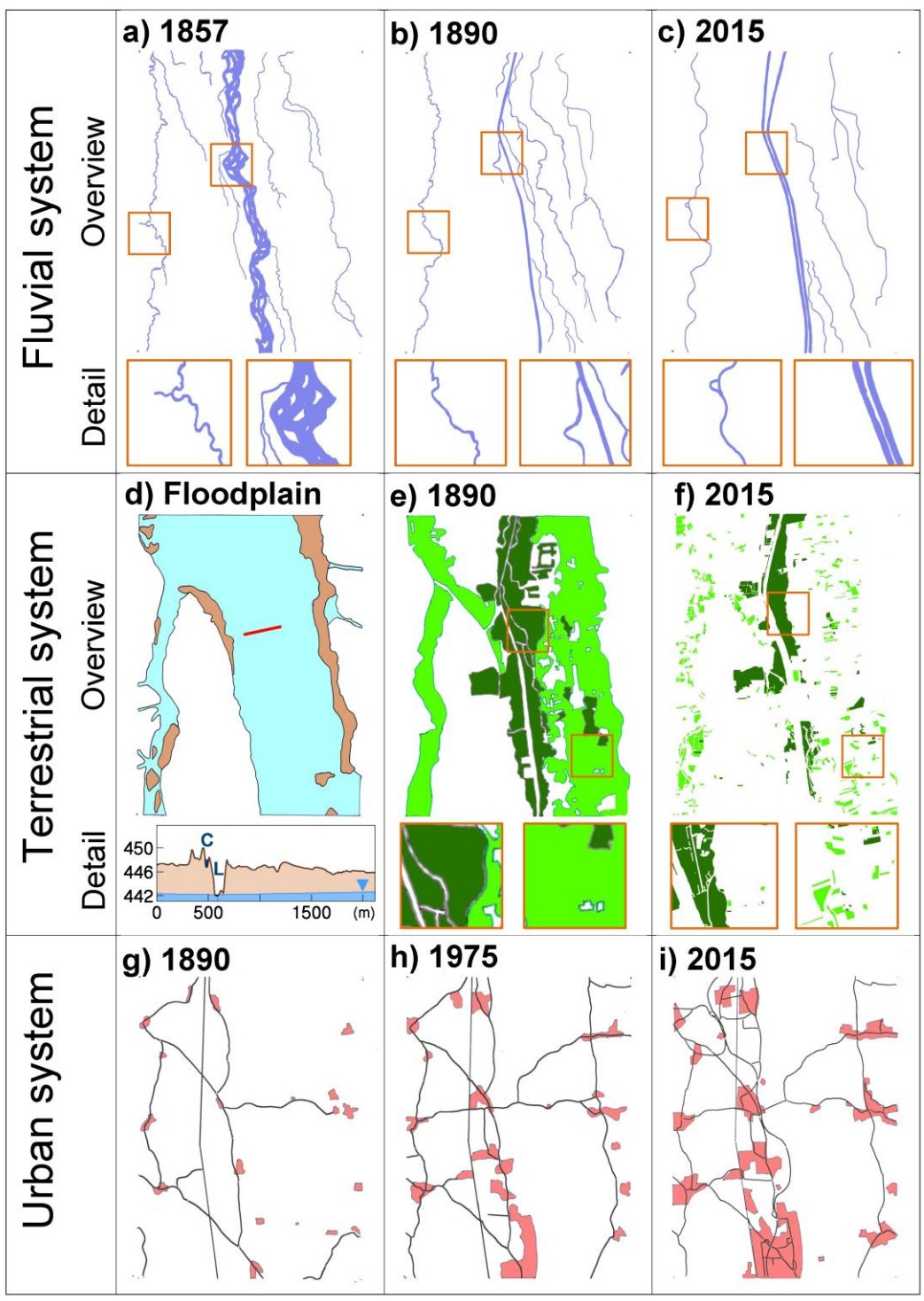

**Figure 2: Change of the fluvial system (a to c), the terrestrial system (d to f) and the urban system (g to i) since the onset of river „training" along the rivers Lech (middle of each panel), Schmutter (left) and Friedberger Ach (right) as extracted from historic maps (numbers denote the year of the respective map and hence usually show the situation a few years earlier). The rivers drain to the north with catchment areas of about 3850 km² for Lech, 360 km² for Schmutter and 450 km² for Friedberger Ach at the northern end. Panel d shows the flood plain with alluvial soils (blue) and remnants of peat soils (brown). The detail panel is a cross section along the red line in the plan view of the floodplain; C and L denote the canal and the river Lech, ▼denotes measured actual groundwater depth at station D36 (www.nid.bayern.de). Panels e and f show riparian forests (dark green) and wet grassland (light green). Panels g to i show towns (red) and major roads and railways. The size of each panel is 9.1 km × 12.2 km. The coordinate of the south-east corner is 48.423° N and 10.940° E.**

## 3. Ecological in-stream consequences

The Anthropocene is characterized by unprecedented rapid loss of biodiversity, with freshwater taxa being particularly affected. The most threatened organisms are typically those that depend on the aquatic or riparian environment for at least part of their life cycle (Fig. 3). Within aquatic habitats, species which are highly specialized and depend on multiple factors for completion of their life cycle are particularly endangered. This holds true for stream fishes (Mueller et al., 2018) as well as for unionid mussels that depend on specific host fish species (Geist, 2011). For a long time, pollution of surface water bodies was considered to be the primary reason for species declines. Because poor water quality also threatened human health, industrialized countries have made large efforts to improve water quality, which has reached high standards again. However, structural changes to most river systems, at least in part attributable to flood protection measures such as levee construction, continue to be a major challenge and often negate improvements in water quality. Globally, habitats associated with 65% of continental water discharge are classified as moderately to highly threatened (Vörösmarty et al., 2010) and in Europe, an average of 60% of protected species and 77% of habitat types are considered to have unfavourable conservation status, with an even higher proportion occurring in rivers, lakes and wetlands (European Environment Agency, 2015). Consequently, urgent action is needed to meet the targets formulated in the European Habitats Directive (Council of the European Communities, 1992) and the Water Framework Directive (Council of the European Communities, 2000), which aim at a "good ecological status or potential".

Rivers are four-dimensional systems because they have longitudinal, lateral, vertical and temporal dimensions (Ward, 1989). Many of their ecosystem functions and services depend on high levels of connectivity among these dimensions throughout entire catchments as well as on dynamic flow regimes (Postel and Richter, 2003). A good example of this is seen in most temperate river systems, where historically there were clear, if complex, linkages of river and floodplain (compare Fig. 2 a and d); these linkages have almost disappeared (compare Fig. 2 c and d). Most riverine species depend on different habitats during their development that need to be linked. For instance, floodplains can provide important rearing habitat for juveniles of specialized fishes such as salmon (Katz et al., 2017). Consequently in-stream restoration measures that do not consider connection with the floodplain are generally insufficient to restore populations of such fishes (Pander and Geist, 2018). The importance of such structural deficits has only been recently recognized. These structural deficits include not only hydraulic decoupling and loss of connectivity of river systems, but also changes to flow regimes and sediment budgets that have major consequences for aquatic biota (Geist and Hawkins, 2016).

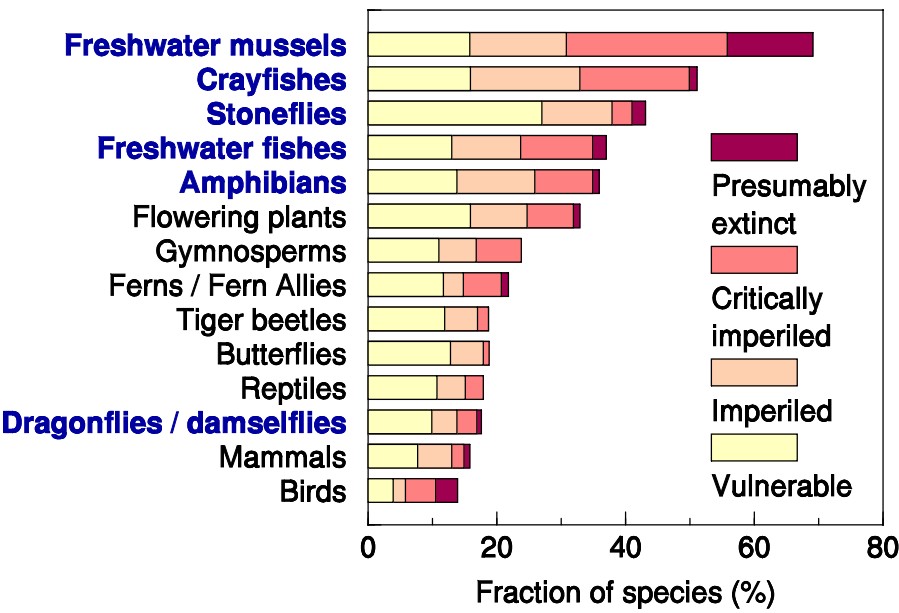

**Figure 3: Share of animal and plant species at risk in the USA (modified from Stein et al., 2000); groups of species that require an aquatic environment during at least part of their life cycle are printed in bold blue. Many declining species in non-aquatic groups depend on riparian habitats.**

Changes to sediment budgets in streams provide a little noticed but major example. Increased fine sediment input into streams results in undesirable ecological consequences such as increased fish egg mortality. Such problems were often uncritically linked to land use and particularly to erosion processes within the catchments. These links fail to explain why this problem has increased during the last decades although erosion has been high since Neolithic times (Dreibrodt et al., 2010; Dotterweich, 2013). They also do not take into account that even small amounts (less than 1% of typical input of fine

sediment) can clog interstitial pore space in stream gravels, making them unsuitable for fish spawning and egg development (Auerswald and Geist, 2018).

The explanation for these sediment effects lies in the break of the natural hydraulic coupling between the river and its floodplain. In natural systems, flooding transports a large proportion of the sediment onto the floodplain where it is deposited behind the natural levees that develop during flooding. The deposition of natural levees is the result of the sharp

decrease in flow velocity of the overtopping water, which reduces sediment transport capacity. The coarsest grain sizes are deposited first and build natural levees (Fig. 4 a). Moreover, water is trapped on the floodplain, where it deposits sediment as alluvial loam and infiltrates into the groundwater. This raises groundwater levels and thus increases backflow through interstitial pore spaces into the river. The backflow flushes interstitial pores that may have become clogged during the preceding hours of the flood, before aging and consolidation of freshly deposited fines hinders resuspension. In contrast,

constructed levees have the opposite effect (Fig. 4 b). These levees keep fine sediments in the river. By increasing flow height, they force sediment-laden flood water to infiltrate into groundwater through interstitial spaces until the interstitial

spaces are clogged, blocking the vital exchange between groundwater and surface water. The same results if rivers and streams are relocated from the bottom of a valley to its edge, e.g. within land consolidation acts that were realized in the majority of central Europe during the second half of the 20[th] century aiming to alleviate cultivation and to optimize agro-economic conditions. Simultaneously regular deposition of sediment on the floodplain is halted. Such sedimentation historically was basis of floodplain fertility and made floodplains the cradle of human civilization over 6000 years ago (Verhoeven and Stetter, 2010).

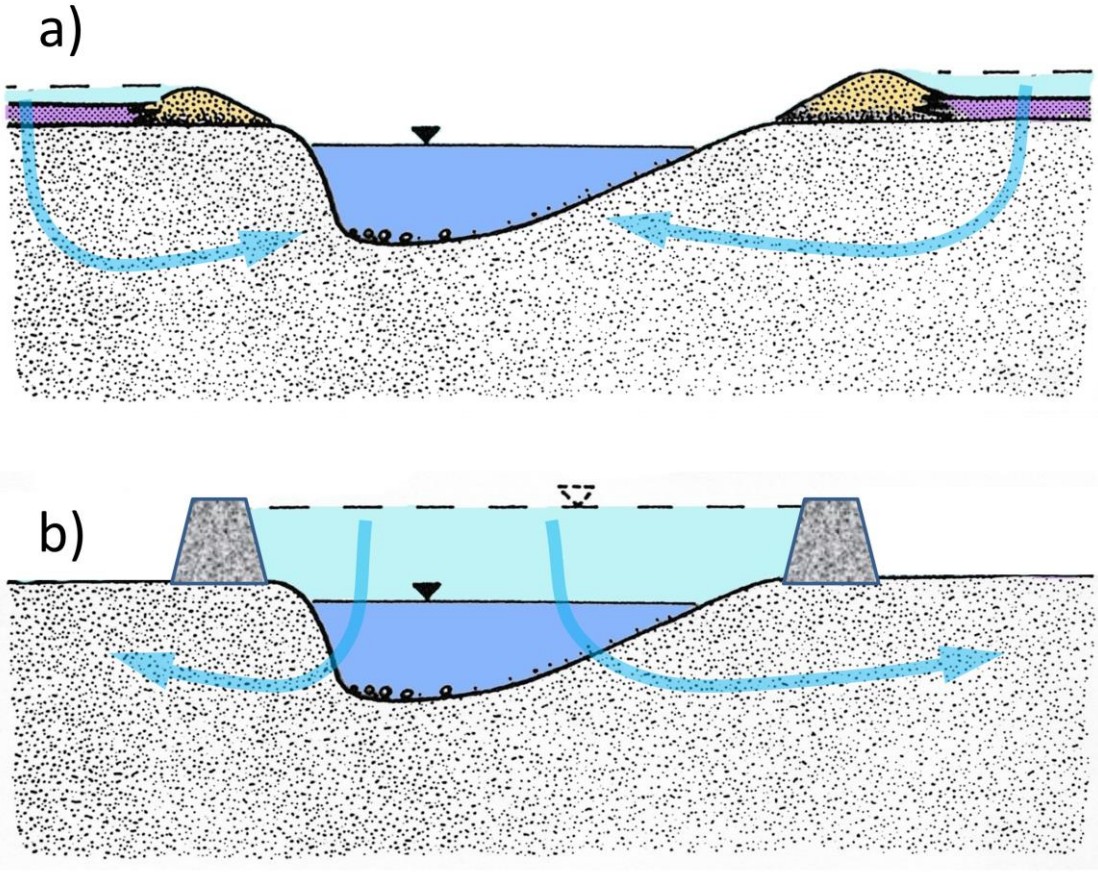

**Figure 4: Comparison of water flows (blue arrows) compelled by natural levees (yellow areas) (top) and by constructed levees (grey structures) (bottom); alluvial loam settles behind natural levees (purple layer).**

Natural river systems are dynamic places where sediment deposited by one flood might be swept downstream by another flood hundreds of years later, supporting forests and fields in the meantime. Functioning floodplains on a large scale remain

today mainly along large tropical rivers, such as the Mekong and the Amazon, although hydropower development is rapidly changing their functionality even there, reducing their value for floodplain agriculture and fisheries (e.g., Ziv et al., 2012).

However, erosion and sedimentation patterns have also changed fundamentally within the river. This is due to the frequent, often concurrent or subsequent to levees, installation of dams, which stabilize the riverbed and offset the natural erosive forces caused by the increased hydraulic radius and bed gradient. In return, dams generate economic value from electric power production, provision of irrigation water and urban water supplies, shipping and the like. When sediment is trapped behind a dam, the river becomes sediment starved below the dam (Kondolf, 1997). This causes unwanted consequences downstream, such as river bed incision and destabilization (Surian and Rinaldi, 2003), threatening bridges and eroding impermeable sediments that protect deeper groundwater layers, i.e. groundwater break-through.

In addition to the changes in erosion and sedimentation patterns and the increased hydraulic decoupling, inundation of the floodplain increases risk of direct contamination of water bodies from settlements (e.g., oil tanks), industrial areas (e.g., chemicals) and agricultural areas (e.g. pesticides, fertilizers, fine sediment) located therein. High runoff from built-up areas, roads and arable fields may additionally deliver substances to water bodies such as road dust, which may contain contaminants (e.g., residues of lubricants), particulate matter (e.g., tire particles) or agrochemicals; this may happen even during moderate rains that do not cause flooding of the main river. On the other hand, if contamination of the river system occurs as in the deadly chemical spill of the River Rhine in 1986 during a fire at a Sandoz warehouse, side arms and channels within a floodplain can act as important refugia for aquatic biota and facilitate faster recolonization.

## 4. Ecological consequences beyond the stream

From an ecosystem perspective, rivers and their former floodplains have become increasingly homogenized, especially where levees or coerced incision are dominant features. There is growing realization that the disconnection of rivers from their floodplains has had major ecological consequences. In addition to the decline in abundances of fishes and mussels (and their fisheries), it has greatly decreased wetlands needed by migratory waterfowl and caused riparian forests and natural retention areas to become small and fragmented (Fig. 2 e, f). Within undisturbed river systems and their floodplains, mosaics of heterogeneous habitat and connectivity govern much of their conservation value. Historically, floodplains were major habitats characterized by a pronounced patchiness of bare or herbaceous or wooded, fertile or infertile, wet or dry places. This patchiness provided the basis for diverse aquatic (Pander et al., 2018) and terrestrial (Krause et al., 2011; Rothero et al., 2016) biological communities. At the same time floodplains were major corridors that connected distant landscapes from alpine areas down to the river mouth through which even large animals like red deer (*Cervus elaphus* L.) could move and pass by urban areas along the river (Wagenknecht, 2000; Amezega et al., 2002).

The construction of levees stopped the flooding that created this habitat patchiness and connectivity while opening opportunities to use these long corridors for major infrastructure such as highways, railroads or even airports (compare Fig. 2 g, h, i). This has resulted in the tremendous extent of transportation infrastructure in alluvial valleys in the USA (Blanton and Marcus, 2009). In densely populated areas of Europe, this problem is even more pronounced, as documented for Switzerland

(Ewald and Klaus, 2009). Such infrastructure has continued to develop despite the EU Environmental Impact Assessment Directive, which obliged member states to conduct mandatory environmental assessments for wetland conversion prior to implementation (Peters and von Unger, 2017). While floodplains formerly were effective corridors for wildlife movement, allowing exchange among many types of habitats and regions, wildlife populations have become fragmented by large infrastructure installation and live today largely in habitat islands (Shepard et al., 2008).

At the same time, construction of levees – often accompanied with lowering of the river bed – allowed draining of wetlands to convert their fertile alluvial soils to cropland at the expense of riparian forests and wet grasslands (Liu et al., 2005) (compare Fig. 2 e and f). Even without additional drainage, the groundwater levels have dropped (by 4 m in Fig. 2 d, detail view) below depths where they can support wet vegetation types with shallow rooting depth. While arable use has been present on the drier parts of the floodplains for millennia, it expanded enormously following the training of river courses

with the help of levees. This expansion started only 200 years ago with the "Rhine corrections" in Europe and has spread on all continents since then (Nienhuis, 2008; Mauch and Zeller, 2008; Nilsson et al., 2005). In particular, traditional land use of wet meadows has almost disappeared. Wet meadows, with their extraordinary floristic and faunistic richness, have become highly endangered habitats (Amezega et al., 2002; European Union, 2016). There is no other vegetation complex that has suffered from loss and biotic degeneration as much as wet grasslands (Ratcliffe, 1984; Schrautzer et al., 1996; Rothereo et

al., 2016; Krause et al., 2012). Overall, the last century has seen half of the world's wetlands lost (Eglington et al., 2008; Dungan, 1993). Although wetland protection is officially a priority since the 1970s for the 170 nations that signed the Ramsar Convention (www.ramsar.org), wetlands continue to be threatened by being drained and reclaimed (Verhoeven and Stetter, 2010).

Another poorly appreciated impact of levees is on carbon and nitrogen sequestration. Wet soils store large amounts of

organic carbon and nitrogen (Wiesmeier et al., 2012), especially when managed as grasslands (Jenny, 1940). Drainage and subsequent cultivation of floodplains protected by levees has released a large share of the formerly sequestered carbon and nitrogen. Estimates show that carbon on the order of 10,000 t km$^{-2}$ and nitrogen on the order of 1,000 t km$^{-2}$ was released into the atmosphere (C) and into the hydrosphere (N) (Van der Ploeg et al., 1999) following such changes. Even higher losses occurred where peatlands were drained (Schothorst, 1977). Peatlands typically developed along the fringes of

floodplains (Fig. 2 d) where the groundwater table is high, ground surface is low and only small amounts of fine sediment reach these distant areas during flooding;. On wide floodplains peatlands may originally have extended over thousands of square kilometres. Lowering the groundwater table (Fig. 2 d, detail view) eliminated the conditions under which peat accumulates and destabilized the peatland, causing land subsidence. Former $CO_2$ sinks have thus turned into $CO_2$ sources. Converting peatlands to cropland has been identified as the most detrimental land use from an atmospheric perspective

(Bryne et al. 2004) and even on nation-scale these areas constitute one of the major sources of $CO_2$ despite their small contribution to total land area. Given that climate change increases the frequency, severity, and extent of floods (Milly et al., 2002, Huntington, 2006), drainage of soils rich in organic matter, which is enabled by levee construction, thus contributes to increasing flood risk by contributing to climate change.

Overall, four factors associated with levees are likely to increase flood risk. Increase in flow velocity and release of carbon to the atmosphere have already been discussed. Here we discuss increased surface runoff and land subsidence. The conversion of land use from grassland to cropland and built-up areas results in a loss of soil buffering capacity due to sealing and the loss of organic matter as well as to decreased rain infiltration and water storage. Thus, surface runoff as predicted by the SCS curve number method (USDA-NRCS 2004) has increased (Van der Ploeg et al., 2000). Applying the approach of Van der
Ploeg et al. (2000) to the floodplain shown in Fig. 2 predicts that the change in land use between 1890 and 2015 increased surface runoff on this floodplain by almost 300%. Runoff from arable land increased by more than 200% while runoff from built-up areas increased by 900%. Half of these increases occurred during the last four decades, suggesting that changes in land use intensify the water cycle to much higher degree than climate change does. Finally, drainage of soils results in shrinkage and organic matter decomposition, causing land subsidence behind levees. Subsidence can be on the order of 1 cm
15  yr$^{-1}$ (Schothorst, 1977; Price et al., 2003) and further increases the need for flood protection.

## 5. Outlook

In the past, there were many good reasons for river reconstruction such as controlling disease through sewage collection and treatment (Preston and Van De Walle, 1978; Nithsdale, 1996; Kesztenbaum and Rosenthal, 2017), hydropower extraction (Koch, 2002), improving navigability (Smith and Winkley, 1996), and reclamation of land for urbanization, infrastructure
and arable agriculture by increasing return periods of floods (Déchamps et al., 1988). Today, however, there is growing realization that complete separation of rivers from their floodplains via levees and coerced river incision has created as many problems as it has solved. Even though most of our examples were derived from Europe and North America, where the development started and has proceeded farthest, human alteration of river and floodplain functioning is not restricted to the Northern Hemisphere. Similar developments can be observed in other regions such as tropical Asia, Africa or South America
where mistakes made in the West are often repeated (Winemiller et al., 2016; Brakenridge et al., 2017; Latrubesse et al., 2017). This development tends to disregard the fact that, especially in the developing and emerging economies of the global south, populations are concentrated in floodplain areas; these floodplains provide important livelihood opportunities but also create large vulnerability to "killer floods" (Kundzewicz et al., 2014). Traditional communities and their economies in the Amazon are well-adapted to flood events, and may demonstrate aspects on how floods can be incorporated in the daily life
of densely populated countries and modern economies (Junk et al., 2011).

Part of the solution involves removing or setting back some levees,  restoring a least a few functional floodplains along major rivers (Malavoi, 1998), and redirecting new housing and other economic development onto lands with less severe

flood risk (Brakenridge et al. 2017). Such approaches may require an elevation of the river bed and translocation of settlements as has happened on the lower River Rhine in the Netherlands; they are thus not popular among politicians, yet they can have great benefits to the majority of people. Alternatively, levees can be set back from the rivers, creating linear floodplains that support wetlands and backwater aquatic habitats. On a larger scale, restored floodplains can be created in

large areas that not only would contain floodwaters, providing relief for downstream levee systems, but would be farmed when not flooded (most years); these farmed floodplains would mainly feature pasture and annual crops such as rice and smaller grains (Suddeth et al., 2016). Farmed floodplains can also serve as seasonal habitat for waterfowl and migratory fishes (Opperman et al. 2017). Such actions, referred to as green infrastructure, can have major benefits not only for people but for the natural world, as diverse managed floodplains become integrated into flood management systems. However,

green infrastructure options can typically only be realized along floodplains that are not yet heavily urbanized. Thus, following the precautionary principle, conservation of the few remaining intact floodplain systems should have greatest priority. The more a floodplain system has been degraded, the more important it becomes to prioritize conservation of the few remaining functionally intact patches. Pristine river floodplains are highly dynamic landscapes, and their biota are *per se* adapted to a certain degree of ecological disturbance. This makes river floodplains relatively easy to restore - at least in

temperate regions - despite the enormous costs that these restoration measures create through the deconstruction of fixed channels, levees, and dams.

In agreement with Di Baldassarre et al. (2013) but expanding their view to include ecological and economic aspects, we propose a transdisciplinary approach to address the interrelated, complex and dynamic social, hydrological, ecological and economic challenges on floodplains. Transdisciplinarity has been promoted as an adequate scientific response to pressing

societal problems even though it is far from being academically established and from being effectively supported by funding and research institutions (Jahn et al., 2012). Transdisciplinarity is understood as a collaboration of academic and non-academic thought styles to break ground for a comprehensive, multi-perspective, common-good oriented trajectory of development (Pohl, 2011). This could guide the interaction of institutions and governance processes with hydrological and ecological processes on floodplains.

## 6. Conclusions

Large floods will always be with us, overwhelming levees and other defences, and creating „disasters" of flooded towns and farms. This realization should result in programs that focus on flood management rather than control. Part of the solution, as indicated by Di Baldassarre et al. (2018), is to develop policies and educational programs that reverse, or at least keep from

growing, the consequences of the levee effect. A comprehensive flood management system should include actions throughout entire catchments, coping for both, small-scale flash floods and large-scale inundations with the goals of improving both socio-economic and ecological conditions. Actions could include taking areas currently behind levees for use

as restored floodplains. Some of these areas could become permanent „wild" floodplain ecosystems, managed mainly for natural features, while others would be farmed, enabling large areas to be zoned as flood relief areas. "Flood-risk management that interweaves structural with nonstructural approaches can keep floods away from people and people away from floods (Opperman et al. 2017, p. 217.)"

Overall, the multiple, interconnected, and often unintended socioeconomic and ecological consequences of traditional flood protection measures must be better considered before planning, construction and restoration of levees and other modifications. Because traditional flood protection requires huge financial investments and results in major and persistent changes to the landscape, it is essential to objectively consider alternative solutions, such as green infrastructure, in an open discussion with stakeholders and policy makers. The discussion must include the full suite of arguments and not ignore

future costs for levee repair, especially following unexpected failures, that typically only accrue decades after construction. Creative flood management, including restoration of functional floodplains and other riverine habitats, can have major positive effects on biodiversity; this in turn can result in provision of ecosystem services that are often erroneously considered to be conflicting targets. In our view, the financial resources available for flood management and provision of ecosystems services, such as biodiversity conservation, can be synergistically combined. This ultimately requires a

transdisciplinary approach that integrates knowledge from ecologists and engineers as well as socio-economists. Emphasis must be put on integrated flood management rather than relying on technical protection measures.

Data availability. No data sets were used in this article. The historical maps are available at https://geoportal.bayern.de/geodatenonline/seiten/bayernatlas-plus_info (last accessed 22 October 2018).

Author contributions. KA and JG conceived, designed and prepared the manuscript together with PM and SPS.

Competing interests. The authors declare that they have no competing conflict of interest.

Acknowledgements. This commentary was inspired by an earlier opinion paper on unintended consequences of structural flood protection by Di Baldassarre et al., 2018. Their focus on socio-economic aspects motivated us to extent the topic on unintended ecological aspects. Our work was supported by the German Research Foundation (DFG) and the Technical University of Munich (TUM) in the framework of the Open Access Publishing Program.

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
