# Peer review of "Hess Opinions: Socio-economic and ecological trade-offs of flood management – benefits of a transdisciplinary approach"

_Hydrology and Earth System Sciences, 2018_

## Referee Comment (RC1) · Anonymous Referee #1 · 26 Nov 2018

This is a nice opinion paper, which discusses the side of effects of structural flood protection. It starts from the recently discussed issue of the safe-development paradox (levee effect) and moves towards a more critical assessment of ecological impacts. The argument is not new, as vast literature is available, but this is a commentary and the main arguments are well supported by the cited literature. Indeed, there is still a major lack of fundamental understanding of these issues, and more transdisciplinary research is needed.

I have two main comments that I hope can help improve this opinion paper. First, I think the paper would benefit from at least a paragraph in which the negative (environmental and social) impacts of structural flood protection are more faily compared to the positive (economical) effects, e.g. growth or development. Second, as the paper suggests a transdisciplinary research agenda, I think that the authors should be aware that Di Baldassarre et al. (2013) published a paper on the same journal (HESS) arguing for transdisciplinarity for a better understanding of deltas and floodplains as human-environment systems. I also suggest a few more references on the topic that might help the revision of this manuscript.

Suggested references

Burton, C. and Cutter, S. L.: Levee failures and social vulnerability in the Sacramento-San Joaquin Delta area, California, Nat. Hazards Review, 2008.

Di Baldassarre, G., Kooy, M., Kemerink, J. S., and Brandimarte, L.: Towards understanding the dynamic behaviour of floodplains as human-water systems, Hydrol. Earth Syst. Sci., 17, 3235-3244, https://doi.org/10.5194/hess-17-3235-2013, 2013.

Jahn, T., Bergmann, M., and Keil, F.: Transdisciplinarity: Between mainstreaming and marginalization, Ecol. Econom., 79, 1–10, 2012.
* * *

---

## Referee Comment (RC2) · Anonymous Referee #2 · 26 Nov 2018

This is a timely paper that provides excellent insight into the complexities of flood-plain use, flood management, the unintended consequences of engineering solutions to flood protection and management. In this regard, the overview of socio-economic and, ecological consequences within and beyond the river provided in Figure 1 was particularly useful. The examples used in the paper also provided good insights, and although these are derived from Europe and North America, they are of global relevance (e.g., the importance of side arms and channels within a floodplain as refugia for aquatic biota in the case of catastrophic events, pg 9).

It is, however, important to note that the impacts of human alteration of river and flood-

plain form and functioning is not unique to the northern hemisphere, from where most of the examples are derived. In many developing and emerging economies in the Global south, populations are concentrated in floodplain areas as these provide important livelihoods opportunities. This increases their vulnerability and "killer floods," mostly affect developing countries (Kundzewics et al. 2013). As economies develop, and the capability for implementing improved flood mitigation improves, the insights and recommendations from Auerswald et al. are relevant not only for conceptualizing proactive mitigation strategies, but also for developing appropriate policy interventions. If they wish, the authors might want to draw on some of the "management rather than control" (pg 11) approaches that are applied in regions where the construction of infrastructure (e.g., levees) is less well established (see Brackenridge et al., 2017). This is especially important for emerging economies where human habitation and associated pressures on floodplain ecosystems are likely to drive investment in quick fix, technological solutions.

Brakenridge et al., 2017. Design with nature: Causation and avoidance of catastrophic flooding, Myanmar, Earth-Science Reviews 165: 81-109, https://doi.org/10.1016/j.earscirev.2016.12.009.

Kundzewicz, Z.W., et al., 2013. Flood risk and climate change: global and regional perspectives. Hydrological Sciences Journal, 59 (1), 1–28.

Minor text edit suggestions.

Pg 9, 10, Revise end of sentence: On the other hand, if contamination of the river system occurs as in the deadly chemical spill of the River Rhine in 1986 during a fire at a Sandoz warehouse, side arms and channels within a floodplain can act as important refugia for aquatic biota and facilitate faster subsequent recolonization

Pg 9, 24, red deer is a specific example – I am not sure if a species name is required here,

---

## Referee Comment (RC3) · Anonymous Referee #3 · 30 Nov 2018

The opinion paper from Auerswald et al. gives a good overview about the ecological and economic consequences of river channelization and rectification, levee building, flood control, river damming, and the consequent loss of ecosystem services to humans. The authors argue that the traditional engineering solutions of river channelization should include alternative solutions that consider more balanced decisions involving both ecological and economic measures (green infrastructure). They conclude that the conservation of the few remaining pristine floodplain systems should have highest priority, and I strongly agree with this argument.

Although the main findings and arguments are not new from the perspective of river

and wetland ecology, I like the opinion paper because it highlights the urgent demand to continuously raise awareness about the economic and ecological consequences of river channelization to the general public, stakeholders, and policy makers. This awareness is particularly lacking in most parts of densely populated central Europe, and partly North America, where most rivers were channelized, dammed, and diked more than a hundred years ago. As such, because most citizens of these countries never lived along a pristine river and floodplain, the ecosystem services that these ecosystems provide are also poorly known and acknowledged by most societies.

As elsewhere, ecologists face strong opposition and drag when trying to conserve pristine river floodplains, or when they make proposals for river and floodplain restoration. In densely populated regions, there are many economic activities along rivers and in floodplains that create complex conflicts of interest, such as between agriculture, forestry, housing and urban development, flood protection, industrial needs and water carriage. I think that these conflicts of interest can only be solved when a sound evaluation of the economic value of ecosystem services (including flood protection) is weighted against the economic return from channelized rivers and destroyed floodplains. I therefore agree with Reviewer 1 that the paper could benefit from the inclusion of some statement where positive and negative economic effects of river channeling are compared. Given the enormous damage and repair costs that densely populated countries increasingly experience due to catastrophic flood (and drought) events, alternative, green infrastructure solutions along rivers and floodplains are the single sustainable way to prevent societies from further damage.

In the main conclusion, however, the paper could also benefit from the inclusion about the opportunities that emerge from the restoration of already channelized river and decoupled floodplain systems. There is an increasing number of river and floodplain restoration projects all over the northern hemisphere, and most of them show that even small-scale projects are able to locally restore pristine conditions, to increase habitat and species diversity, and to restore further (but not all) important ecosystem
services, such as water retention and flood control. Pristine river floodplains are highly dynamic landscapes, and their biota is per definition adapted to a certain degree of ecological disturbance. This makes river floodplains relatively easy to restore - at least in temperate regions - despite the enormous costs that these restoration measures cause through the deconstruction of channel fixation, dikes, and dams.

Some suggestions for potential inclusion in the reference list:

-A couple of years ago, the French concept of the "Espace de liberté" for rivers was developed through Malavoi J.-R. (1998) – Bassin Rhône-Méditerranée-Corse. Guide Technique N°2 : Détermination de l'espace de liberté des cours d'eau. Secrétariat Technique du SDAGE, Lyon, 40 p. It describes the idea to deconstruct river fixations to increase ecosystem services provided by free-flowing river channels.

-As most rivers and floodplains in the northern hemisphere are strongly modified through humans, important ecological concepts (such as the flood-pulse concept by Junk et al. 1989) mostly derive from other parts of the world, such as the Amazon. Traditional communities and their economies in the Amazon are well-adapted to flood events, and this might be a good example how floods can also be incorporated in the daily life of densely populated countries and modern economies. See: Junk WJ, et al. (eds.): Amazonian Floodplain forests: Ecophysiology, Biodiversity and Sustainable Management. Ecological Studies 210, Springer Verlag, Heidelberg, Berlin, New York

-A good review on the impact of river dams for the Amazon is from Latrubesse EM, et al. (2017): Damming the rivers of the Amazon basin. Nature 546: 363-369

---

## Short Comment (SC1) · 7 Dec 2018

**Replies to Reviews #1, #2 and #3 on "Hess Opinions: Socio-economic and ecological trade-offs of flood management – benefits of a transdisciplinary approach" by Karl Auerswald et al.**

We appreciate the encouraging comments and helpful amendments. In blue we explain how we considered the reviewers' advice in our manuscript.

**Anonymous Referee #1**

This is a nice opinion paper, which discusses the side of effects of structural flood protection. It starts from the recently discussed issue of the safe-development paradox (levee effect) and moves towards a more critical assessment of ecological impacts. The argument is not new, as vast literature is available, but this is a commentary and the main arguments are well supported by the cited literature. Indeed, there is still a major lack of fundamental understanding of these issues, and more transdisciplinary research is needed.

I have two main comments that I hope can help improve this opinion paper. First, I think the paper would benefit from at least a paragraph in which the negative (environmental and social) impacts of structural flood protection are more faily compared to the positive (economical) effects, e.g. growth or development.

We added at the beginning of our outlook:
"In the past, there were many good reasons for river reconstruction such as controlling disease through sewage collection and treatment (Preston and Van De Walle, 1978; Nithsdale, 1996; Kesztenbaum and Rosenthal, 2017), hydropower extraction (Koch, 2002), improving navigability (Smith and Winkley, 1996), and reclamation of land for urbanization, infrastructure and arable agriculture by increasing return periods of floods (Déchamps et al., 1988) "

Second, as the paper suggests a transdisciplinary research agenda, I think that the authors should be aware that Di Baldassarre et al. (2013) published a paper on the same journal (HESS) arguing for transdisciplinarity for a better understanding of deltas and floodplains as human-environment systems. I also suggest a few more references on the topic that might help the revision of this manuscript.

Suggested references
Burton, C. and Cutter, S. L.: Levee failures and social vulnerability in the Sacramento- San Joaquin Delta area, California, Nat. Hazards Review, 2008.

Di Baldassarre, G., Kooy, M., Kemerink, J. S., and Brandimarte, L.: Towards understanding the dynamic behaviour of floodplains as human-water systems, Hydrol. Earth Syst. Sci., 17, 3235-3244, https://doi.org/10.5194/hess-17-3235-2013, 2013.

Jahn, T., Bergmann, M., and Keil, F.: Transdisciplinarity: Between mainstreaming and marginalization, Ecol. Econom., 79, 1–10, 2012.

We appreciate drawing our attention to this point and we regret that we only mentioned transdisciplinarity in the Abstract and in the Conclusions without developing this idea further. We added a paragraph on transdisciplinarity at the end of the Outlook.

"In agreement with Di Baldassarre et al. (2013) but expanding their view to include ecological and economic aspects, we propose a transdisciplinary approach to address the interrelated, complex and dynamic social, hydrological, ecological and economic challenges on floodplains. Transdisciplinarity has been promoted as an adequate scientific response to pressing societal problems even though it is far from being academically established and from

being effectively supported by funding and research institutions (Jahn et al. 2012). Transdisciplinarity is understood as a collaboration of academic and non-academic thought styles to break ground for a comprehensive, multi-perspective, common-good oriented trajectory of development (Pohl, 2011). This could guide the interaction of institutions and governance processes with hydrological and ecological processes on floodplains."

**Anonymous Referee #2**

This is a timely paper that provides excellent insight into the complexities of floodplain use, flood management, the unintended consequences of engineering solutions to flood protection and management. In this regard, the overview of socio-economic and, ecological consequences within and beyond the river provided in Figure 1 was particularly useful. The examples used in the paper also provided good insights, and although these are derived from Europe and North America, they are of global relevance (e.g., the importance of side arms and channels within a floodplain as refugia for aquatic biota in the case of catastrophic events, pg 9).

It is, however, important to note that the impacts of human alteration of river and floodplain form and functioning is not unique to the northern hemisphere, from where most of the examples are derived. In many developing and emerging economies in the Global south, populations are concentrated in floodplain areas as these provide important livelihoods opportunities. This increases their vulnerability and "killer floods," mostly affect developing countries (Kundzewics et al. 2013). As economies develop, and the capability for implementing improved flood mitigation improves, the insights and recommendations from Auerswald et al. are relevant not only for conceptualizing proactive mitigation strategies, but also for developing appropriate policy interventions. If they wish, the authors might want to draw on some of the "management rather than control" (pg 11) approaches that are applied in regions where the construction of infrastructure (e.g., levees) is less well established (see Brackenridge et al., 2017). This is especially important for emerging economies where human habitation and associated pressures on floodplain ecosystems are likely to drive investment in quick fix, technological solutions.

Brakenridge et al., 2017. Design with nature: Causation and avoidance of catastrophic flooding, Myanmar, Earth-Science Reviews 165: 81-109, https://doi.org/10.1016/j.earscirev.2016.12.009.

Kundzewicz, Z.W., et al., 2013. Flood risk and climate change: global and regional perspectives. Hydrological Sciences Journal, 59 (1), 1–28.

We added:
"Even though most of our examples were derived from Europe and North America, where the development started and has proceeded farthest, human alteration of river and floodplain functioning is not restricted to the Northern Hemisphere. Similar developments can be observed in other regions such as tropical Asia, Africa or South America where mistakes made in the West are often repeated (Winemiller et al., 2016; Brakenridge et al., 2017; Latrubesse et al., 2017). This development tends to disregards the fact that, especially in the developing and emerging economies of the global south, populations are concentrated in floodplain areas; these floodplains provide important livelihood opportunities but also create large vulnerability to "killer floods" (Kundzewicz et al., 2013). Traditional communities and their economies in the Amazon are well-adapted to flood events, and demonstrate how floods can be incorporated in the daily life of densely populated countries and modern economies (Junk et al., 2011)."

Minor text edit suggestions.

Pg 9, 10, Revise end of sentence: On the other hand, if contamination of the river system occurs as in the deadly chemical spill of the River Rhine in 1986 during a fire at a Sandoz warehouse, side arms and channels within a floodplain can act as important refugia for aquatic biota and facilitate faster subsequent recolonization

We changed the sentence as proposed

Pg 9, 24, red deer is a specific example – I am not sure if a species name is required here,

Species name was added

**Anonymous Referee #3**

The opinion paper from Auerswald et al. gives a good overview about the ecological and economic consequences of river channelization and rectification, levee building, flood control, river damming, and the consequent loss of ecosystem services to humans. The authors argue that the traditional engineering solutions of river channelization should include alternative solutions that consider more balanced decisions involving both ecological and economic measures (green infrastructure). They conclude that the conservation of the few remaining pristine floodplain systems should have highest priority, and I strongly agree with this argument.

Although the main findings and arguments are not new from the perspective of river and wetland ecology, I like the opinion paper because it highlights the urgent demand to continuously raise awareness about the economic and ecological consequences of river channelization to the general public, stakeholders, and policy makers. This awareness is particularly lacking in most parts of densely populated central Europe, and partly North America, where most rivers were channelized, dammed, and diked more than a hundred years ago. As such, because most citizens of these countries never lived along a pristine river and floodplain, the ecosystem services that these ecosystems provide are also poorly known and acknowledged by most societies. As elsewhere, ecologists face strong opposition and drag when trying to conserve pristine river floodplains, or when they make proposals for river and floodplain restoration. In densely populated regions, there are many economic activities along rivers and in floodplains that create complex conflicts of interest, such as between agriculture, forestry, housing and urban development, flood protection, industrial needs and water carriage. I think that these conflicts of interest can only be solved when a sound evaluation of the economic value of ecosystem services (including flood protection) is weighted against the economic return from channelized rivers and destroyed floodplains. I therefore agree with Reviewer 1 that the paper could benefit from the inclusion of some statement where positive and negative economic effects of river channeling are compared. Given the enormous damage and repair costs that densely populated countries increasingly experience due to catastrophic flood (and drought) events, alternative, green infrastructure solutions along rivers and floodplains are the single sustainable way to prevent societies from further damage.

We added at the beginning of our outlook:
"In the past, there were many good reasons for river reconstruction like improvement of disease control by sewage collection and treatment (Preston and Van De Walle, 1978; Nitsdale, 1996; Kesztenbaum and Rosenthal, 2017), hydropower extraction (Koch, 2002),

navigability (Smith and Winkley), and reclamation of land for urbanization, infrastructure and arable agriculture by increasing return periods of floods (Déchamps et al., 1988). "

**In the main conclusion, however, the paper could also benefit from the inclusion about the opportunities that emerge from the restoration of already channelized river and decoupled floodplain systems**. There is an increasing number of river and floodplain restoration projects all over the northern hemisphere, and most of them show that even small-scale projects are able to locally restore pristine conditions, to increase habitat and species diversity, and to restore further (but not all) important ecosystem services, such as water retention and flood control. **Pristine river floodplains are highly dynamic landscapes, and their biota is per definition adapted to a certain degree of ecological disturbance. This makes river floodplains relatively easy to restore - at least in temperate regions - despite the enormous costs that these restoration measures cause through the deconstruction of channel fixation, dikes, and dams**.

Some suggestions for potential inclusion in the reference list:
-A couple of years ago, the French concept of the "Espace de liberté" for rivers was developed through
Malavoi J.-R. (1998) – Bassin Rhône-Méditerranée-Corse. Guide Technique N2 : Détermination de l'espace de liberté des cours d'eau. Secrétariat Technique du SDAGE, Lyon, 40 p. It describes the idea to deconstruct river fixations to increase ecosystem services provided by free-flowing river channels.

-As most rivers and floodplains in the northern hemisphere are strongly modified through humans, important ecological concepts (such as the flood-pulse concept by Junk et al. 1989) mostly derive from other parts of the world, such as the Amazon. Traditional communities and their economies in the Amazon are well-adapted to flood events, and this might be a good example how floods can also be incorporated in the daily life of densely populated countries and modern economies. See:
Junk WJ, et al. (eds.): Amazonian Floodplain forests: Ecophysiology, Biodiversity and Sustainable Management. Ecological Studies 210, Springer Verlag, Heidelberg, Berlin, New York

-A good review on the impact of river dams for the Amazon is from
Latrubesse EM, et al. (2017): Damming the rivers of the Amazon basin. Nature 546: 363-369

We added:
"Pristine river floodplains are highly dynamic landscapes, and their biota are *per se* adapted to a certain degree of ecological disturbance. This makes river floodplains relatively easy to restore - at least in temperate regions - despite the enormous costs that these restoration measures cause through the deconstruction of channel fixation, dikes, and dams."

And additionally cited among others: Malavoi (1998), Latrubesse et al. (2017) and Junk et al. (2011)